# Association Between Circulating Vitamin K Levels, Gut Microbiome, and Type 1 Diabetes: A Mendelian Randomization Study

**DOI:** 10.3390/nu16223795

**Published:** 2024-11-05

**Authors:** Samuel De La Barrera, Benjamin De La Barrera, Marc-André Legault, Isabel Gamache, Despoina Manousaki

**Affiliations:** 1Research Center of the Sainte-Justine University Hospital, Université de Montréal, Montreal, QC H3T 1C5, Canada; samuel.de.la.barrera@umontreal.ca (S.D.L.B.); benjamin.delabarrera@live.concordia.ca (B.D.L.B.); marc-andre.legault.1@umontreal.ca (M.-A.L.); isabel.gamache@umontreal.ca (I.G.); 2Faculty of Pharmacy, Université de Montréal, Montreal, QC H3T 1J4, Canada; 3Departments of Pediatrics, Biochemistry and Molecular Medicine, Université de Montréal, Montreal, QC H3T 1C5, Canada

**Keywords:** mendelian randomization, vitamin K, type 1 diabetes, GWAS

## Abstract

Background/Objectives: Nutritional deficiencies have been proposed as possible etiological causes for autoimmune diseases, among which type 1 diabetes (T1D). Vitamin K (VK) has potentially positive effects on type 2 diabetes, but its role on T1D in humans remains largely unknown. We aimed to examine the presence of a causal association between VK and T1D using a Mendelian randomization (MR) approach. Methods: Genetic variants from a genome-wide association study (GWAS) for VK (*N* = 2138 Europeans) were used as instruments in our two-sample MR study to investigate whether circulating VK levels are causally associated with the risk of T1D in a large European T1D GWAS cohort (18,942 cases/520,580 controls). Through a multivariable MR (MVMR), the effects of both VK and specific gut microbiota on T1D were investigated given that the gut microbiome synthesizes VK. Results: We found that changes in levels of circulating VK did not affect T1D risk in our univariate two-sample MR, but this study had limited power to detect small effects of VK (OR for T1D of less than 0.8). However, our MVMR indicated a suggestive association of VK with the risk of T1D adjusting for two different gut microbiome populations. Conclusions: In conclusion, VK levels are unlikely to significantly affect the risk of T1D, but small effects cannot be excluded, and the role of gut microbiome in this association should be further investigated.

## 1. Introduction

Type 1 diabetes (T1D) is an autoimmune disease, caused by the destruction of the insulin-producing beta cells of the pancreas [1], leading to insulin deficiency, chronic hyperglycemia, and increased morbidity and mortality. The etiology of T1D is multifaceted, involving a complex interplay of genetic variations and environmental factors that disrupt the function of the immune system [2]. Among these factors, nutritional deficiencies, such as in vitamins A, B, C, D, and E, have been proposed as possible etiologic factors for T1D [3]. Recently, vitamin K has gained attention for its potential effects on type 2 diabetes in adults, but its role on the auto-immune T1D remains largely unknown [1,4].

While vitamin K (VK) has historically been recognized for its role in blood coagulation and as an antioxidant and anti-inflammatory agent, recent studies suggest its potential involvement in glucose homeostasis [1]. Vitamin K 1 (VK1), or phylloquinone, obtained from leafy greens and fruits, and vitamin K 2 (VK2), or menaquinone, synthesized by gut microbiota or fermented foods, are the primary forms of VK [5]. Observational studies have linked reduced blood levels of VK to type 2 diabetes (T2D), implicating its role in glycemic regulation and insulin sensitivity [4,6]. However, whether VK is causally related to diabetes and its impact on T1D remains elusive. T1D-induced mice and rat models have shown VK’s potential protective effects against hyperglycemia and diabetes-related complications [7,8]. VK may influence insulin secretion and pancreatic beta cell proliferation or regeneration, mechanisms relevant to T1D pathophysiology [1,9]. Moreover, VK’s transcriptional activity and its conversion between VK1 and VK2 in extrahepatic tissues and the gut microbiome underscore its potential implications in autoimmune processes [10,11,12,13,14]. For instance, it has been suggested that VK acts as an agonist of steroids and xenobiotic nuclear receptors [12,13], explaining VK’s possible involvement in autoimmune diseases such as multiple sclerosis [15]. In addition, the microbiome may convert VK1 to VK2 in the lower intestine in humans, and the gut microbiome also processes VK2 [11], while Mendelian randomization has shown a possible causal association between the gut microbiome and T1D [16]. However, no randomized controlled trials have studied the effects of VK supplementation in humans on the risk of developing T1D.

Collectively, the evidence from the aforementioned epidemiological studies faces challenges in establishing causal associations between VK and T1D due to inherent limitations, such as unmeasured confounding and reverse causation [1,4,5,6,10,11,14,17,18,19,20]. Mendelian randomization (MR) has emerged as a powerful tool leveraging genetic variants associated with biomarkers to investigate causal relationships of modifiable exposures with disease outcomes. Unlike observational studies, MR minimizes biases from confounding and reverse causation by using germline genetic variants randomly assigned at conception [21,22,23,24,25,26,27,28,29]. MR has been used previously to investigate the causal role of various biomarkers in T1D [24,27].

In this study, we aimed to test whether circulating VK1 levels are causally associated with the risk of T1D using two-sample MR. Given the role of the gut microbiome in VK2 synthesis and T1D and the tight homeostasis between VK1 and VK2, we applied multivariable MR to study the potential effect of specific gut microbiome populations on the association between VK and T1D. To do this, we leveraged data from the only available circulating VK1 GWAS in Europeans [30], a large gut microbiome GWAS [31], and the largest available European GWAS on T1D [32].

## 2. Materials and Methods

### 2.1. Overview of the MR Study

The flowchart of our study appears in Figure 1. The direct acyclic graph of our main MR study appears in Figure 2. To analyze the causal effect of VK1 (exposure) on T1D (outcome) using MR, we first identified single nucleotide polymorphisms (SNPs) as instrumental variables (IVs) for serum VK1 (phylloquinone) in a GWAS by Dashti et al. [30]. This GWAS provided summary-statistic results from a European GWAS meta-analysis of the CHARGE Consortium Nutrition Working Group, which totalled 2138 individuals [30]. They performed three different models in which VK1 levels were adjusted for different covariates (Model 1: age- and sex-adjusted model; Model 2: triglyceride-adjusted model; Model 3: vegetable intake-adjusted model). Units of measurement for VK1 were expressed in standard deviations following a natural logarithmic transformation. We retrieved the effects of SNPs associated with VK1 levels on T1D from the largest European T1D GWAS by Chiou et al. (*N* = 18,942 patients of European ancestry and 520,580 controls from 9 European cohorts) [30,32] (Appendix A).

We then aimed to study the potential mediating effect of vitamin K2 (VK2, menaquinone) on the association between VK1 and T1D. Due to the unavailability of a VK2 GWAS, we used the gut microbiome, which synthesizes VK2, as a proxy [5]. An MR study by Luo et al. [16] has already shown a causal effect of four gut microbiome species (class Bacteroidia, Eubacterium eligens, Bacteroidetes, and order Bacteroidales) on T1D; thus, we decided to investigate if these gut microbiome species were mediators or could modify the effect in the association between VK1 and T1D. To do this, we performed multivariable Mendelian randomization (MVMR) using data from a human gut microbiome GWAS by Kurilshikov et al. [31]. Appendix A provides additional details on each GWAS dataset used in our MR analyses.

### 2.2. Three MR Assumptions and Instrumental Variable (IV) Selection

Univariable two-sample MR studies were performed to explore the causality of VK1 on the risk of T1D. The first MR assumption (relevance assumption) requires the IVs to have a strong association with the exposure. To satisfy this assumption, we selected SNPs associated with VK1 levels in the exposure GWAS (*p*-value ≤ 5 × 10^−6^) and had an F-statistic > 10, implying a strong instrument. The F-statistic was computed with the formula $F=\frac{\frac{R^{2\;}}{k}}{\left(1-R^{2}\right)\left(n-k-1\right)}\;$ where k is the number of SNPs and n is the cohort size. The R^2^ is calculated using the formula: $R^{2}=2\times \beta^{2}\times maf\times \left(1-maf\right)$, where β and maf denote the allele effects and minor allele frequency, respectively. Out of the 11 available SNP-IVs in the exposure GWAS, we selected independent SNPs using a clumping approach. We used the LDpair tool from the NIH Ldlink website (https://ldlink.nih.gov) to identify SNP pairs with an r^2^ > 0.1. We retained the most significant SNP from each pair (i.e., the SNP with the lowest p-value in the VK1 GWAS), resulting in 6 SNP-IVs remaining for our MR analysis (Appendix A).

The second assumption (independence assumption) requires that a SNP-IV is not linked to confounding factors that connect the exposure to the outcome. The most common violation of this assumption is when ancestry acts as a confounder, which was accounted for by ensuring that both the exposure and outcome GWAS were performed in populations of European descent. Violation of the second assumption is known as horizontal pleiotropy. The third assumption (exclusion restriction assumption) requires that the IV affects the outcome only through exposure; otherwise, horizontal pleiotropy may be present. To ensure the absence of horizontal pleiotropy, we investigated whether SNP-IVs could be pleiotropic and therefore needed to be removed. To do this, we performed a Phenome-Wide Association Study (PheWAS) on the SNP-IVs using GWASatlas (https://atlas.ctglab.nl/PheWAS, accessed on 2 May 2024) [33]. If a SNP-IV was significantly associated with autoimmune diseases or immunity-related traits (*p* < 1 × 10^−5^), it was considered pleiotropic and was subsequently removed from the analysis (Appendix A). Cohorts from both the exposures and outcome consist of non-Hispanic White individuals, thereby limiting potential confounding effects.

### 2.3. Mendelian Randomization Analysis

We performed univariate MR studies using the TwoSampleMR R package (v0.5.7) [34], applying its default parameters to harmonize the effects of the SNP-IVs between the exposure (VK1) and the outcome (T1D) GWAS. In the primary analysis, we calculated inverse variance weighted (IVW) MR estimates by meta-analyzing all Wald ratios representing the MR effect of each individual SNP-IV. Each Wald ratio was weighted according to the inverse of its variance, with the regression intercept set to zero [35]. We considered an IVW MR *p*-value < 0.05 as statistically significant.

Further sensitivity analyses were conducted to assess the potential presence of heterogeneity and pleiotropy within the VK1 SNP-IVs. We employed four pleiotropy robust MR methods (weighted median, weighted mode, MR-Egger and MR-PRESSO) to evaluate whether the IVW estimates may be biassed by pleiotropy. MR-Egger was used to test the exclusion restriction MR assumption by identifying and adjusting for directional pleiotropy, allowing for an intercept that deviates from zero, with a significant deviation indicating directional pleiotropy [36]. The weighted median method generated median-based estimates, which are valid when less than 50% of SNP-IVs are pleiotropic, while the weighted mode method enabled accurate assessment even when the majority of SNP-IVs are pleiotropic [37]. MR-PRESSO identified outlier SNP-IVs via its global test, then recalculated estimates excluding these outliers. This analysis was implemented using the MR-PRESSO R package (v1.0) [38]. We also calculated Cochran’s Q statistic and its p-value to evaluate heterogeneity among SNP-IVs. Reverse causation (i.e., the possibility that T1D influences VK1 levels rather than the reverse) was assessed using the Steiger directionality test to confirm the causal direction of our association [39].

The power in our main MR analyses was computed using the online tool “mRnd’ (https://shiny.cnsgenomics.com/mRnd/, accessed on 21 July 2024) for a binary outcome. The settings consisted of an alpha level of 0.05, the proportion of cases from the T1D GWAS [32], an OR of T1D compatible with the average MR OR from our IVW analyses, and the R^2^ (proportion of variance explained by the VK1 by its SNP-IVs) corresponding to each MR analysis. We also calculated the minimum MR OR to obtain a power of 0.8.

### 2.4. Multivariable MR (MVMR) Analysis for VK1 and Gut Microbiota

In our MVMR analysis, we sought to investigate the direct effects on T1D risk of VK1 adjusting for four specific gut microbiome bacterial populations. Data for this second exposure were derived from the Kurilshikov et al. GWAS [31]. The Kurilshikov et al. GWAS provides SNPs associated with various types of heritable microbiota taxa and their occurrence and abundance. The four gut microbiome bacterial populations selected based on previously reported causal associations with T1D in a MR study [16] were class Bacteroidia, Eubacterium eligens, Bacteroidetes, and order Bacteroidales. The portion of the VK1 (exposure 1) effect on T1D explained by each of these species (exposure 2) was tested in MVMR as implemented in the MVMR R package version 0.4 [40], which computed adjusted IVW estimates for each of the exposures. We considered class Bacteroidia and order Bacteroidales as a single entity due to the fact that these two species had identical GWAS summary statistics. The *p*-value cut-off for a significant IVW result in our MVMR was defined following a Bonferroni correction as 0.05/3 = 0.017, given that we tested three sets of exposures (i.e., combinations of VK1 with each of the three microbiota species).

## 3. Results

### 3.1. Association Between Vitamin K Levels and Type 1 Diabetes

We evaluated the potential causal association of VK1 with T1D in univariate MR analyses using 6 independent SNP-IVs and their GWAS effects on the exposure (VK1 levels based on three different GWAS models correcting for different covariates). In all three models, all six SNPs had an F-statistic above 10, suggesting that they are strong instruments (Appendix A). Our PheWAS search did not reveal GWAS associations of the VK SNP-IVs suggestive of horizontal pleiotropy (Appendix A).

Our IVW analyses did not reveal any significant MR associations between VK1 and T1D, and pleiotropy robust methods yielded similar null results (Appendix A, Figure 3). As shown in Appendix A, the intercept of the MR-Egger regression provided no evidence of unbalanced horizontal pleiotropy in any of the MR studies, suggesting that the SNP-IVs do not influence T1D through alternative pathways. The non-significant Cochran’s Q *p*-values obtained from both IVW and the MR-Egger methods indicated the absence of heterogeneity among the SNP-IVs, supporting the consistency of the causal estimates. The Steiger directionality test did not suggest the presence of reverse causation in our MR studies, implying that T1D risk does not impact VK1 levels.

In our main MR study, we observed a power of 20%, 16%, and 17% for the first, second, and third VK1 models, respectively, to detect an OR of 0.93, which represents the average of the three obtained IVW MR ORs. Our MR analyses had a statistical power of 80% to show an OR below 0.8 (or above 1.2) for T1D per SD change in serum VK1 level. This suggests that while our study can reasonably exclude the presence of a VK1 effect on T1D with an OR < 0.8 (or >1.2), it may not have sufficient statistical power to detect smaller effects of VK1 on T1D. Specifically, the low power observed in detecting the effect at the average OR of 0.93 suggests that effects of subtle changes in VK1 levels might not be captured by our analysis. The results of our power analysis are displayed in Appendix A.

### 3.2. MVMR Testing Effect of Vitamin K Levels and Gut Microbiota on Type 1 Diabetes

To test mediation or effect modification of the association between VK1 and T1D by gut microbiota, we performed an MVMR using as exposures serum VK1 levels (based on the three GWAS models) and each of the three gut microbiome population categories: Eubacterium eligens, Bacteroidetes, and order Bacteroidales/class Bacteroidia. For this analysis, only five VK1 SNP-IVs were used, since one was not present in the gut microbiome GWAS.

The IVW results from the MVMR analyses are presented in Appendix A. Our IVW analyses yielded suggestive results for two out of the three gut microbiome populations using SNP-IVs from the first VK1 GWAS model (IVW *p*-values 0.03 and 0.04 for Bacteroidetes and order Bacteroidales/class Bacteroidia, respectively). However, the other two models did not show significant associations, which may be attributed to their lower statistical power, as indicated by our MR power analysis. None of these estimates remained significant after the Bonferroni correction. Our results suggest a possible causal effect of VK1 on T1D risk conditioning on specific gut microbiome populations.

The MR-STROBE checklist with the methods and findings of our MR study appears in Appendix A.

## 4. Discussion

In this study, we applied an unbiased MR approach to study the causal association between serum VK1 level and T1D. Based on observational studies on the effects of VK on T1D, we hypothesized that our MR approach could confirm a causal association between VK and TID. Our two-sample MR analyses did not reveal a significant effect of circulating VK1 levels on the risk of developing T1D. However, our MR study was unable to exclude small effects of VK1 on T1D (OR < 0.8 or >1.2), but our findings can reasonably exclude larger effects, indicating that altering VK1 levels would not have a strong impact on T1D development. We therefore conclude that the effects described in previous studies are likely driven by unmeasured residual confounders, and supplementation in VK1 is unlikely to prevent T1D.

Several observational studies have shown that VK2 might have a preventative effect on some diabetes-associated comorbidities. While most of these studies link the effects of circulating VK2 to type 2 diabetes risk [1,4,8], blood levels of VK1 have also been shown to decrease the risk of some T1D-related complications such as nephropathy and cellular stress in mouse models [7,9]. Therefore, VK1 could be more strongly associated with T1D-related comorbidities rather than with the risk of T1D development per se. This hypothesis merits further investigation using MR studies using future large GWAS for T1D complications and circulating VK1 levels.

Due to the lack of a VK2 GWAS, we sought an alternative approach to test the effects of VK2 on the association between VK1 and T1D by conducting an MVMR using gut microbiota as proxies for VK2. Indeed, the selected bacterial populations were previously shown to be causally associated with the risk of T1D in a MR study [16]. This approach was chosen because VK2 is synthesized by the gut microbiome, and there is evidence suggesting that VK1 may be converted into VK2 by the gut microbiome in humans [5,11]. Regarding the mechanism of the conversion, it has been suggested that in animals, VK1 is cleaved by the microbiome to form an intermediate called menadione or VK3. VK3 is then absorbed and converted into VK2 in tissues [10]. The results of our MVMR suggest a possible effect of VK1 conditioning on specific gut microbiota species. This aligns with the prior MR study that established a causal association between certain gut microbiota species and T1D [16]. Additionally, it supports the hypothesis that VK may influence gut microbiota population dynamics and community composition, potentially affecting the risk of developing T1D [41,42]. However, the absence of a significant effect of VK1 on T1D in our univariate MR makes the interpretation of this result challenging. A possible explanation could be that the gut microbiome represents a potential collider in the association between VK1 and T1D rather than a mediator of this association (Figure 4). Colliders are variables that are independently affected by the exposure and the outcome [43], and their presence can induce spurious associations or alter the association’s observed magnitude (Figure 4) [43]. Performing an MVMR using SNP-IVs from both exposures (i.e., VK1 and gut microbiota populations) could help in the interpretation of the results of our MVMR analysis (where only SNP-IVs from VK1 were used). However, full summary statistics from the VK1 GWAS are not available, which precludes undertaking such analysis. Also, it is important to note that humans have a different VK metabolism and gut microbiome compared to animals, and there is a limited understanding of the mechanisms behind the VK1 to VK2 conversion [11,20,44]. There is currently no previous literature identifying or suggesting that the gut microbiome may act as a collider in the causal association between VK1 and the risk of T1D or other autoimmune diseases. Further investigation is needed to confirm the exact role of the human gut microbiome in the conversion of VK1 into VK2 and its impact on the association between VK and T1D.

We acknowledge several key limitations in our study. Our objective was to investigate the effect of VK1 and VK2 on T1D development. However, given the absence of GWAS for VK2, we used gut microbiota as a proxy to test the impact of VK2 in the association between VK1 and T1D. Although a GWAS for VK1 was available, its limited sample size reduced the statistical power of our MR analyses, thereby constraining our ability to detect small causal effects. The generation of larger GWAS datasets for VK1 and VK2 is essential to robustly evaluate the causal relationships of these vitamins with T1D using MR. Additionally, the traditional MR approach cannot assess non-linear effects, preventing us from evaluating whether only extreme levels of VK1 could influence T1D or if interactions exist between the microbiome and VK1. All GWAS datasets used in this MR analysis are derived from European cohorts, which precludes the generalisability of our findings to other ancestries. Finally, this study is the first of its kind to explore the interplay between VK, the gut microbiome, and T1D. As a result, comparison with previous literature is limited, as existing studies either focus on narrower aspects of these associations or lack relevant context in relation to our findings. This highlights the importance of further research to clarify the connections among these three players.

## 5. Conclusions

Our two-sample MR study indicates that serum VK1 levels do not have a large causal effect on the risk of T1D. When we factor in gut microbiome populations, we observe a suggestive causal effect, which should be interpreted with caution and merits further investigation. Overall, our study suggests that VK1 may not be a viable preventive target for T1D to test in future studies.

## Figures and Tables

**Figure 1 nutrients-16-03795-f001:**
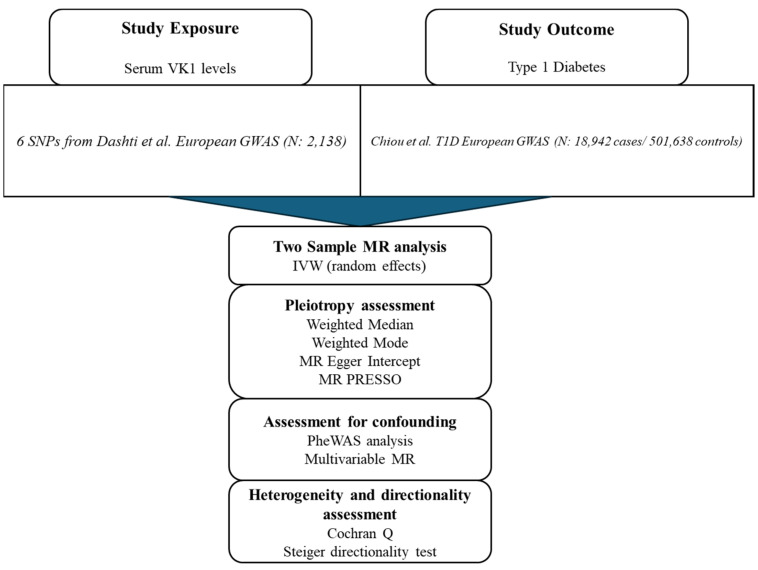
Flowchart with the design of our MR study.

**Figure 2 nutrients-16-03795-f002:**
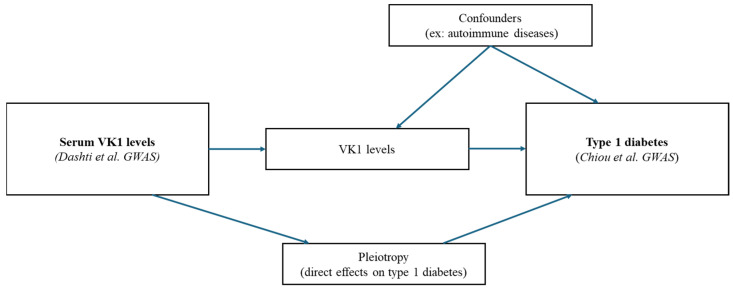
MR direct acyclic graph (DAG) of our study.

**Figure 3 nutrients-16-03795-f003:**
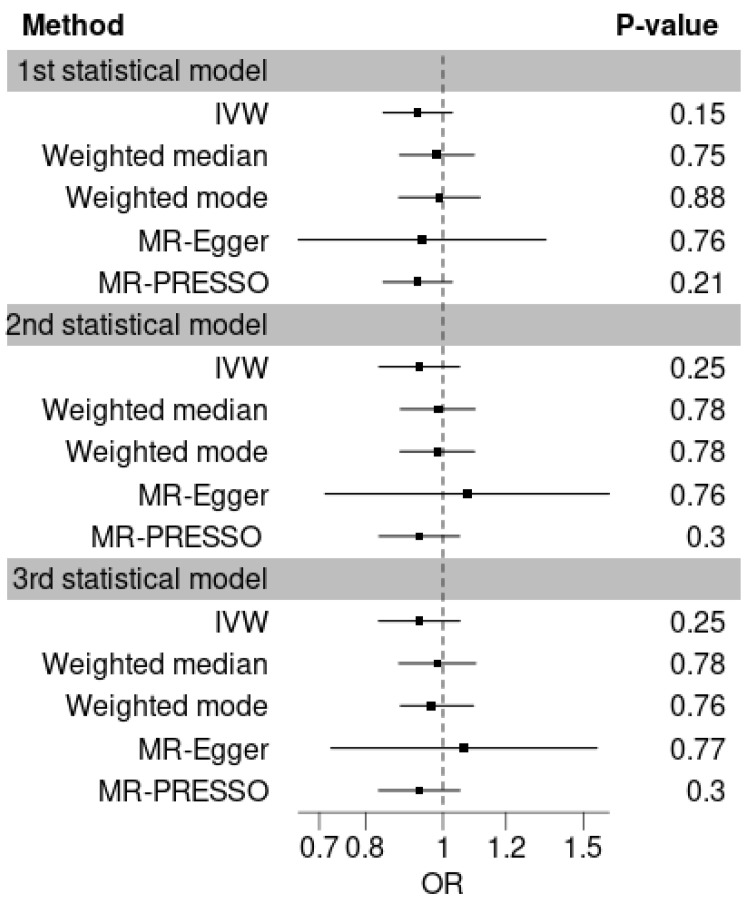
Forest plots with the results of the main MR analysis. The results represent the MR OR for T1D per SD increase in circulating VK1 levels measured according to three GWAS models and using various MR methods.

**Figure 4 nutrients-16-03795-f004:**
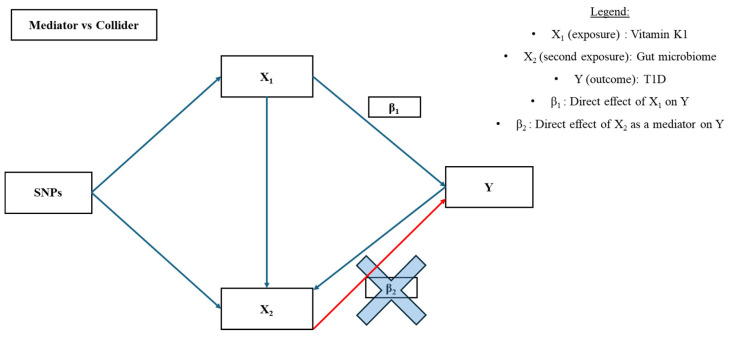
Direct acyclic graph (DAG) of our MVMR analysis, including gut microbiome as a second exposure. If exposure 2 (gut microbiome) is a mediator, then the red line (effect β_2)_ shows the indirect effect of exposure 1 (VK1) on the outcome Y (T1D). If exposure 2 (gut microbiome) is a collider (blue line between Y and X2), then accounting for the β_2_ can induce a spurious association between exposure 1 (VK1) and the outcome Y (T1D).

## Data Availability

All GWAS data used are publicly available through the GWAS catalog: https://www.ebi.ac.uk/gwas/home accessed on 10 June 2023. The full GWAS statistics of the SNP-IVs for VK1 can be found directly from the Dashti et al. publication; summary-level data for this GWAS are not publicly available. All other data supporting the reported results can be found in the article or the Appendix A.

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
