# Peer review of "Association Between Circulating Vitamin K Levels, Gut Microbiome, and Type 1 Diabetes: A Mendelian Randomization Study"

_nutrients, 2024, doi:10.3390/nu16223795_

Round 1
Reviewer 1 Report
Comments and Suggestions for Authors
Review of the article Nutrients- 3257252
1. For the main question addressed by the research. The first aim of the study is to test whether circulating Vitamin K1 levels are causally associated with the risk of type 1 diabetes (T1D) using a two sample Mendelian randomization study and the second is the potential effect of specific gut microbiome (acting as a proxy for vitamin K2) populations on the association between vitamin K and T1D What parts do you consider original or relevant for the field? Mendelian randomization studies are a reliable source of information in the purpose of determining associations between a risk factor and a health outcome being less influenced by confounders that might affect the outcome.
2. For specific gaps in the field, it does in the paper address. The role of K vitamin in auto-immune T1D. To date there are no randomized controlled trials that study the effect of vitamin K supplementation in humans and the risk of developing T1D.
3. For the subject area compared with other published material. Large Mendelian randomization studies allow to evaluate associations between risk factors and different health outcomes reducing the effect of possible confounders.
4. For specific improvements the authors should consider the methodology. For further controls should be considered. Given the available data sets methodology is scientifically correct.
5. Please describe how the conclusions are or are not consistent with the evidence and arguments presented. Please also indicate if all main questions posed were addressed and by which specific experiments. Conclusions are consistent with data and respond to the aims of the study, limitations are acknowledged and suggested also future research.
6. For the references appropriate. Yes
7. For any additional comments on the tables and figures and quality of the data. - No other comments
Author Response
Thank you for taking the time to review this manuscript.
Comments:
- For the main question addressed by the research. The first aim of the study is to test whether circulating Vitamin K1 levels are causally associated with the risk of type 1 diabetes (T1D) using a two sample Mendelian randomization study and the second is the potential effect of specific gut microbiome (acting as a proxy for vitamin K2) populations on the association between vitamin K and T1D What parts do you consider original or relevant for the field? Mendelian randomization studies are a reliable source of information in the purpose of determining associations between a risk factor and a health outcome being less influenced by confounders that might affect the outcome.
2. For specific gaps in the field, it does in the paper address. The role of K vitamin in auto-immune T1D. To date there are no randomized controlled trials that study the effect of vitamin K supplementation in humans and the risk of developing T1D.
3. For the subject area compared with other published material. Large Mendelian randomization studies allow to evaluate associations between risk factors and different health outcomes reducing the effect of possible confounders.
4. For specific improvements the authors should consider the methodology. For further controls should be considered. Given the available data sets methodology is scientifically correct.
5. Please describe how the conclusions are or are not consistent with the evidence and arguments presented. Please also indicate if all main questions posed were addressed and by which specific experiments. Conclusions are consistent with data and respond to the aims of the study, limitations are acknowledged and suggested also future research.
6. For the references appropriate. Yes
7. For any additional comments on the tables and figures and quality of the data. - No other comments
Response:
We sincerely appreciate and thank the positive evaluation of the reviewers.
Reviewer 2 Report
Comments and Suggestions for Authors
In this interesting Mendelian randomization study by De La Barrera et al., the association between circulating vitamin K levels and the risk of T1D was investigated. Additionally, through a multi-variable Mendelian randomization, the potential effect of the gut microbiome on the association between vitamin K and T1D was examined.
This study essentially refutes the presence of an unbiased association between Vitamin K1 and the risk of T1D. However, it reveals the role of the gut microbiome as a potential collider in the association between VK1 and T1D. Therefore, this study reinforces the need for a better understanding of the role of the gut microbiome in autoimmune diseases and the pathogenesis of T1D.
The paper is overall well-presented, the discussion is broad and exhaustive, and the figures and tables are clear.
A few minor suggestions from my side:
- Since the role of the gut microbiome has also been considered in the analysis, I suggest slightly modifying the title to make the paper more intriguing. A possible suggestion could be: "Association between circulating Vitamin K levels, gut microbiome, and Type 1 Diabetes: A Mendelian randomization study."
- In the introduction section, please provide a reference for the following statement: “Recently, vitamin K has gained attention for its potential effects on type 2 diabetes in adults, but its role in autoimmune T1D remains largely unknown.”
- Please clarify why the study did not require IRB approval
Author Response
Thank you for taking the time to review this manuscript. Please find our detailed responses below, along with the revisions in track changes in the re-submitted files.
Comments 1: Since the role of the gut microbiome has also been considered in the analysis, I suggest slightly modifying the title to make the paper more intriguing. A possible suggestion could be: "Association between circulating Vitamin K levels, gut microbiome, and Type 1 Diabetes: A Mendelian randomization study."
Responses 1: We thank the reviewer for this valuable feedback. Following your suggestion regarding our manuscript's title, we have considered your recommendation and modified the title to "Association between circulating Vitamin K levels, gut microbiome, and Type 1 Diabetes: A Mendelian randomization study."
Comments 2: In the introduction section, please provide a reference for the following statement: “Recently, vitamin K has gained attention for its potential effects on type 2 diabetes in adults, but its role in autoimmune T1D remains largely unknown.”
Responses 2: In the introduction section, we have added references to support the statement regarding vitamin K’s potential, which are the references:
- Karamzad, N., et al., A systematic review on the mechanisms of vitamin K effects on the complications of diabetes and pre‐diabetes. BioFactors, 2020. 46(1): p. 21-37.
- Ho, H.-J., M. Komai, and H. Shirakawa, Beneficial Effects of Vitamin K Status on Glycemic Regulation and Diabetes Mellitus: A Mini-Review. Nutrients, 2020. 12(8): p. 2485.
Comments 3: Please clarify why the study did not require IRB approval
Responses 3: We have clarified in the manuscript that this study did not require IRB approval, as all data used was derived from publicly available summary statistics.
- L. 355-357: This study did not require specific IRB approval, since no individual-level data were used. All GWAS cohorts contributing data to this study received ethics approval from their respective ethics review board.
Reviewer 3 Report
Comments and Suggestions for Authors
This study is aimed to examine the presence of a causal association between Vitamin K and T1D using a Mendelian Randomization approach.
Therefore, it is expected that it will be able to provide important information for preventing or treating type I diabetes.
However, correction and supplementation of the following matters are required.
Overall, the research plan is well designed and well summarized in the attached table.
However, I wish a more detailed explanation was provided for readers' understanding.
1. The research method requires a detailed explanation of the detailed methods applied. For example, two sample MR analysis, Pleiotrophy assessment, Assesment for confounding, Heterogeneity and directinality assessment etc.
2. In the discussion process to interpret the results, a more detailed explanation and comparison with various previous studies are required.
Author Response
Thank you for taking the time to review this manuscript. Please find the responses below and the corresponding revisions in track changes in the re-submitted files.
Comments 1: The research method requires a detailed explanation of the detailed methods applied. For example, two sample MR analysis, Pleiotrophy assessment, Assesment for confounding, Heterogeneity and directinality assessment etc.
Response 1: We appreciate the reviewer's suggestion for a more detailed explanation for our research methods. As requested, we have added more information on the research methods and provided a clearer interpretation of the results in the results section (lines 112-116,129-145, 152, 156-178, 206, 212, 229-232,244-256).
Comments 2: In the discussion process to interpret the results, a more detailed explanation and comparison with various previous studies are required.
Response 2: We appreviate the suggestion to compare our findings with previous studies. However, to our knowledge, this is the first manuscript evaluating this relationship in humans and the interplay between VK, the gut microbiome and T1D. Therefore, no direct comparisons could be included due to the lack of previous studies. We have added this specification at lines 281-282 and 300-302. Despite this, we provided more insight on our MVMR results elaborating on how VK might influence gut microbiome populations which might influence T1D risk. We added this at lines 283-285.
Round 2
Reviewer 3 Report
Comments and Suggestions for Authors
Overall, I think the revisions and supplements have been done well.